# Oral Care Cards as a Support in Daily Oral Care of Frail Older Adults: Experiences and Perceptions of Professionals in Nursing and Dental Care—A Qualitative Study

**DOI:** 10.3390/ijerph19159380

**Published:** 2022-07-31

**Authors:** Jessica Persson, Isabelle Johansson, Cristina Joy Torgé, Eva-Karin Bergström, Catharina Hägglin, Inger Wårdh

**Affiliations:** 1Department of Health Sciences, University West, 461 86 Trollhättan, Sweden; 2Centre for Gerodontology, Public Dental Service, Region Västra Götaland, 402 33 Gothenburg, Sweden; eva-karin.bergstrom@vgregion.se (E.-K.B.); catharina.hagglin@gu.se (C.H.); 3Department of Periodontology, Institute of Odontology, Sahlgrenska Academy, University of Gothenburg, 405 30 Gothenburg, Sweden; 4Institute of Gerontology, School of Health and Welfare, Jönköping University, 553 18 Jönköping, Sweden; joy.torge@ju.se; 5Department of Cariology, Institute of Odontology, Sahlgrenska Academy, University of Gothenburg, 405 30 Gothenburg, Sweden; 6Department of Behavioral and Community Dentistry, Institute of Odontology, Sahlgrenska Academy, University of Gothenburg, 405 30 Gothenburg, Sweden; 7Department of Dental Medicine and Academic Centre of Geriatric Dentistry, Karolinska Institute, 141 52 Huddinge, Sweden; inger.wardh@ki.se; 8Department of Health Sciences, Karlstad University, 651 88 Karlstad, Sweden

**Keywords:** frail older adults, interprofessional collaboration, nursing care, oral health, oral care documentation, person-centered care, qualitative methods

## Abstract

Frail older adults often have poor oral health. In Sweden, oral care cards are designed to be used as an interprofessional tool for documenting the oral health status of older adults with extensive care needs and to describe oral care recommendations. The aim of this study was to explore nursing and dental professionals’ experiences and perceptions of oral care cards. Nursing and dental care staff were interviewed in groups or individually. The recorded data were transcribed verbatim and analyzed using qualitative content analysis. A theme emerged: Navigating an oral care responsibility that is not anchored in the nursing and dental care context. The theme was elucidated in three categories: “Accessibility and usefulness”, “Coordination between nursing and dental care”, and “Ethical approach”. The participants perceived a lack of surrounding frameworks and collaboration concerning oral care and the use of oral care cards. An oral care card could ideally facilitate interprofessional and person-centered oral care. However, oral health does not seem to have found its place in the nursing care context. Further research is needed to investigate how oral care cards ought to be developed and designed to support oral health care work.

## 1. Introduction

The United Nations Sustainable Development Goal 3 in the 2030 Agenda is to “*ensure healthy lives and promote well-being for all at all ages*”. It emphasizes the need for support systems in healthcare that consider an aging population with complex needs of care [1].

Adults in long-term care (LTC) or home care services are among the frailest groups in society, with severe risk for oral diseases [2]. Ageing generally involves physical changes that make it more difficult to maintain oral health [3]. In turn, poor oral health can have negative consequences on other areas of health. Cardiovascular diseases [4], respiratory diseases [5], diabetes [6], and malnutrition [7,8], among other conditions, are correlated with oral diseases. Poor oral health also affects the quality of life of older adults by causing pain, reduction of chewing ability, and lowered well-being and self-esteem [9,10]. Poor oral hygiene and dental caries [11] is common among home care recipients [2] as well as residents in LTC, especially in people with neurocognitive disorders [12].

The literature shows that in many countries, the oral care of older adults receiving old age care services is insufficient [2,11]. Nursing staff report a perceived lack of knowledge regarding oral health and oral care [13]. Interventions, including oral health education for nursing staff, have been carried out with mainly positive results, but there is no established evidence in the area due to varying design and quality of the studies [14,15,16]. Nevertheless, recommendations addressed to nursing staff such as oral care checklists [17], picture-based oral procedure cards, oral care plans, and toothbrushing protocols and schedules, may have positive effects on the oral hygiene of older adults in LTC [15].

In Sweden, a dental care remuneration program was implemented in 1999, with the goal of ensuring oral health and quality of life and nutrition for frail older adults. The program entitles older adults with extensive care needs to subsidized dental care, together with an annual oral assessment at home (own home or nursing home) by a dental hygienist, free of charge. With the ambition to include oral care in nursing care and to promote learning regarding oral care, oral assessments were to be made with nursing care staff present [18].

To facilitate oral health care work by nursing staff, the program also launched the use of a paper-based oral care card to be completed after each oral assessment. The oral care card contains information on the person’s oral status, individual recommendations for oral hygiene routines, oral care products and procedures, and the ability of the older person to manage oral hygiene independently or with help [18]. Originally, the oral care card had a standardized appearance, but adjustments have been made over the years in different administrative regions and by private operators of dental care.

In 2019, more than 91,000 persons in Sweden were issued an oral care card after receiving an oral assessment [19]. However, the use and follow-up of the oral care cards has some reported shortcomings. For instance, there were reports pointing out how completed oral care cards were misplaced and the recommended oral hygiene products were not available [20]. This indicates hindrances for the oral care cards to be used for their intended function. For the oral care card to be used as an interprofessional and interorganizational tool for better oral health, there is a need to highlight involved professionals’ perspectives. The aim of this present study was to explore nursing and dental professionals’ experiences and perceptions of oral care cards.

## 2. Materials and Methods

### 2.1. Design

This is a qualitative study with exploratory design and an inductive approach. Group and individual interviews were performed using a semi-structured interview guide.

### 2.2. Participants

The study was performed in Region Västra Götaland, in the southwest of Sweden. Different professions that were considered to contribute with different perspectives and have knowledge of oral care cards were identified, and purposeful sampling was used to recruit participants in different roles in nursing and dental care. For a variety of participants, recruitment was made in different parts of the region in order to include both urban and rural areas. The participants were of active working ages (20–65 years).

The interviewed participants consisted of nursing assistants who work in LTC or home care services, dental hygienists that perform oral assessments for oral care cards, and persons in different leadership positions in nursing and dental care. Three nursing assistants worked as oral care aides with some responsibility for oral health care work at their wards [21] and were interviewed in a separate group due to their special knowledge and experiences of the oral care cards. In another group interview, a dental hygienist and a strategist in municipal health care were interviewed together, as they belonged in an already existing collaborative intervention in the municipality [22].

The data collection continued until an adequate amount of information had been collected with regard to the study aim, which resulted in 10 individual or group interviews. In total, 30 participants were interviewed. Table 1 shows an overview of the participants.

The participants received written and verbal information about the study, voluntariness, and confidentiality before starting the interviews. All participants gave their signed informed consent.

### 2.3. Data Collection

Prior to the study, two pilot interviews (one with a dental hygienist and one with a registered nurse) were performed by the first author (JP), who had experience from interviewing in earlier projects. The pilot interviews were not included in the analysis because of a close collegial relationship to the interviewer. The interview guide was adapted with questions concerning structural and organizational aspects of the use of the oral care card. These changes were used in the subsequent interviews, particularly for the interviews with participants in leadership positions.

Seven group interviews and three individual interviews were performed between October 2020 and June 2021. Due to the risk for spread of the COVID-19 infection, a great part of the interviews was conducted digitally using the communication platform Teams or the programme Skype. The interviews were recorded by Dictaphone. All interviews were conducted by the same moderator, JP, and at group interviews, IJ or CH acted as observers. All interviews were conducted in Swedish. The quotes in the result section were translated to English and controlled by all authors to ensure that the essence of the quotes was correct.

To stimulate the discussions, different oral care cards were sent to the participants before the interviews. The interviews started with the question “Could you please tell me about your experiences using oral care card?”, followed by questions about the function, design, and usefulness of the oral care cards. The interviews lasted between 28 and 61 min (average 40 min) and were recorded and transcribed verbatim by JP and IJ.

### 2.4. Analysis

The transcribed data were distributed to the research group to individually read through the data and obtain a sense of the whole. Qualitative content analysis was conducted [23,24] by JP and IJ and the analysis was discussed among members of the research group.

Data were divided into meaning units, and condensed meaning units were coded. Codes with similar content were sorted into subcategories. Subcategories with similar content were interpreted and abstracted into categories. The authors discussed and compared the categories with the transcribed data throughout the entire process until all authors came to a consensus on the results. Table 2 presents an example from the analysis regarding the category “Coordination between nursing and dental care”.

## 3. Results

The theme “Navigating an oral care responsibility that is not anchored in the nursing and dental care context” is hereby presented through two representative quotes. One participant from nursing care expressed:

Nurse assistant A, group 1: “*Well, the idea with these cards—Are they meant to be used for something?”*

A dental hygienists echoed this:

Municipal dental hygienist, group 7: “*Who looks at the oral cards and does anybody read them? What are they for?”*

The participants described a situation where the oral care cards were not established in the routines and structures of nursing and dental care, nor did they regard the autonomy of the older adults. The theme was abstracted from three categories: “Accessibility and usefulness”, “Coordination between nursing and dental care”, and “Ethical approach”. They are, together with their subcategories, presented in Table 3.

### 3.1. Accessibility and Usefulness

The category “Accessibility and usefulness” describes the participants’ experiences regarding information on the oral care card and their relevance. They describe a need for adapting the oral care card for all users regardless of profession—from dental hygienists completing the oral care cards to nursing staff using them in daily work. The subcategories are “Need of overview of dental status”, “Need of description of oral care needs”, “Need of customization for different users”, and “Need of oral care card to be within reach”.

#### 3.1.1. Need of Overview of Dental Status

The participants were quite satisfied with design and content of the oral care cards, usually visually showing teeth and tooth replacements such as prosthesis, bridges, and implants. Visualization was considered helpful since nursing personnel expressed a lack of knowledge about what to find in the mouth.

Registered nurse: “*I kind of want to know how it looks like in the person*’*s mouth. It*’*s hard for me to tell if something is one or the other kind of dental bridge. It*’*s beyond my knowledge*, *to be honest*.”

A nurse assistant described how the oral status of new residents was unknown until oral assessment was performed and the oral care card was issued, which sometimes could take several months.

#### 3.1.2. Need of Description of Oral Care Needs

Residents in nursing care have an individual care implementation plan, developed by nursing staff together with the older adults and/or relatives describing how the care should be delivered. The registered nurse pointed to a need to include oral care in this individual plan. Nurse assistants confirmed that the oral care cards serve the purpose of describing oral care needs and recommended products and providing information on the older adults’ need for assistance with oral care.

Nurse assistant A, group 2: “*It’s important to know about the person’s dental implants since they require special care. With implants, you have an interdental brush. If one has a dry mouth, that’s also important information*.”

In home care service, the care implementation plan is available in the staff’s digital planning tool. The staff in home care services believe that they could be allocated time to support older adults’ oral care needs if they reported these needs to the municipality case worker. However, with no allocated time, the staff were not able to act.

#### 3.1.3. Need of Customization for Different Users

Oral care cards need to be useful for different involved professions in both nursing and dental care. The nursing staff wanted the oral care card to be easy to interpret, not least if they do not have Swedish as their first language.

Nurse assistant B, group 4: “*If there is too much text*, *you don’t bother to look at it. So*, *there should be more pictures than text* … *condense the text to a minimum*, *I mean*.”

Participants from dental care reflected that older adults and their formal and informal carers needed customized instructions of how to clean the mouth, preferable with image support or photography. The registered nurse wanted details regarding the perspective of nursing, such as the possibility to prioritize the recommendations.

#### 3.1.4. Need of Oral Care Card to Be within Reach

The participants wanted the oral care card to be easily accessible. Participants from dental care recommended it to be placed on the wall in the bathroom next to the sink, but some nursing staff considered this placement a violation of the integrity of the older adults. In many cases, the oral care card was placed in a binder together with other information about the person, such as medications. One nurse assistant reflected upon this:

Nurse assistant B, group 1: “[I need] *the necessary information to be able to help out when I’m standing there* [helping with oral care], *instead of running to look at binders. Uh, is it their intention for us to bring the whole binder into* [the bathroom]?”

Nursing staff in home care services received most of their information in a smart phone application, and never went into bathrooms unless the digital planning tool told them they have a task there. The dental hygienists thought that the oral care card could be easily administered through a digital solution.

### 3.2. Coordination between Nursing and Dental Care

This category describes the complexity surrounding the oral care card and the interorganizational and interprofessional context where it is placed. The data formed four subcategories: “Unstructured handover”, “Uncertain distribution of work and responsibilities”, “Lack of specific follow-up”, and “Non-integrated systems”.

#### 3.2.1. Unstructured Handover

The dental hygienists, the LTC manager, and nursing staff all described the handover of the oral care card information as lacking established routines. The registered nurse described a situation where she received secondhand information that reached her two weeks late and orally, almost by coincidence.

A dental hygienist expressed her concerns regarding the complicated and uncertain process in writing and distributing the oral care cards, and she feared the work was “meaningless”.

Dental hygienist A, group 6: “*The nursing staff are very seldom there during an oral assessment. So directly when I finish the assessments*, *I try to contact someone in the staff and inform them about which recommendations we gave and to whom*.”

Interviewer: “*With whom do you discuss that*?”

Dental hygienist A: “*Yeah*, *well the first best nursing assistant in the unit. Depends on who’s available*.”

The dental hygienists had different strategies to compensate for the lack of handover: writing brief notes with the most important information, leaving printed cards for recommended oral care products, and gathering all nursing staff to ensure that as many as possible heard the information.

#### 3.2.2. Uncertain Distribution of Work and Responsibilities

The registered nurse described how the oral care card could support the nurse’s needs concerning work and responsibilities of oral care, which at the time for the interview was experienced as uncertain. Clarity in the responsibilities of involved staff from nursing and dental care was sought, as well as the role of professionals in leading positions.

Registered nurse: “*I want to know what our task is. What is my role*? *I want to be informed when we need to do something that requires getting a prescription*. *Or if* [the dental staff] *feel that they want to follow up on something*.”

#### 3.2.3. Lack of Specific Follow-Up

The dental hygienists expressed that sometimes the oral care card was nowhere to be found at the persons’ homes or had never been included in the nursing care records, perhaps because the oral care card is not defined as a relevant document to integrate into documentation. The dental hygienists also expressed the need to compare current oral health status with previous years to identify and predict the correct progression and risk of oral disease for the older adults.

Dental hygienist B, group 6: “*It is important to be able to compare current oral status with the previous years. Say if the patient has got three more root remnants* [since the last time], *then the patient has a big problem there*.”

Some of the participants from nursing care also stated the importance of follow-ups. The LTC manager described how other licensed professions, such as physiotherapists and dietitians, had regular follow-ups concerning different treatments, technical aids, and care, but that dental care did not follow up on the recommendations given.

#### 3.2.4. Non-Integrated Systems

Several nursing staff participants expressed how oral care was a part of daily self-care but there was seldom anyone who could describe how and when it was performed. Nursing staff in home care services described the oral care as invisible. In general, there was a lack of consensus regarding the purpose of the oral care card, and dental hygienists prized its value more than those in nursing care.

Municipal dental hygienist, group 7: “*For us from dental care*, *it is an important record. The municipality should also give it the same importance—these are our recommendations*, *something important that we want to share. Not a piece of paper* (…) *to be put in a pile of other oral care cards*.”

With no instructions regarding oral care, the nursing staff described that they base their oral care work from general recommendations from their own dental visits.

### 3.3. Ethical Approach

This category focuses on aspects regarding the older adults that are not a part of the current oral care card, such as the general health status and autonomy.

The sub-categories are “Need to regard the state of the older adult”, “Need to respect the older adult’s preferences”, and “Need to promote participation”.

#### 3.3.1. Need to Regard the State of the Older Adult

The nursing staff mentioned that the general health status of the older person was not described on the oral care card, even though this highly affects the person’s ability to perform oral care. They wished that the card could provide customized advice on oral care with regard to the older person’s varying health status.

Nurse assistant A, group 2: “*For a person with a dementia diagnosis, it varies from day to day what they can and cannot do. It often has to do with how that person feels and*, *as in other somatic conditions*, *the day-to-day form*.”

A dental hygienist described that she had the responsibility to perform oral assessments at ten to twelve LTC facilities and that it was impossible to take the time to remember all general health statuses of the older adults.

#### 3.3.2. Need to Respect the Older Adult’s Preferences

The nursing staff highlighted the importance of considering the autonomy of the older adults. A majority of the older adults had performed their own daily oral care for decades. One nurse assistant reflected:

Nurse assistant C, group 1: “*I want to get information about which patients keep their protheses at night*, *and which ones I should remove them for. There are a few that don’t want to take them out at night. That information ought to be written down somewhere*.”

The oral assessments performed by the dental care staff are made during a limited time. A dental hygienist stated that she had issued the recommendations on the oral care card, sometimes without having expressly secured consent from the older adult. If dental hygienists recommend daily assisted oral care, but the older adult opposes this, it is important to document this for further care planning.

#### 3.3.3. Need to Promote Participation

The nursing staff discussed the need for letting the older adults participate in their own daily oral care and to take autonomy into consideration when issuing a new oral care card. They highlighted that recommendations concerning oral care needed to be adapted to the older adults’ ability to perform and maintain the function of self-care.

Nurse assistant B, group 2: “*I think that one should give them the possibility to do oral care themselves*, *in the level of their capacity. We shouldn*’*t take that away from them so they can participate in taking care of their teeth. And we can support them in that*.”

## 4. Discussion

### 4.1. Discussion—Results

Oral care cards describe individual daily oral care needs for frail older adults who often have an increased risk for oral diseases [11]. For nursing and dental care staff, it is also a summary of the person’s present oral status and a document to be used interprofessionally and interorganizationally with the aim to ensure an oral health perspective in healthy ageing [18]. The results of this present study are based on interviews with nursing and dental care professionals working in nursing care, and a broad picture of the strengths and limitations of oral care cards has emerged.

The theme is in line with a fundamental problem that has been highlighted in studies for many years regarding oral health in nursing care: oral health and oral care have not yet found their place in the nursing care context [25,26,27,28,29,30]. Although the oral care card was in use, the practice surrounding it was detached from the day-to-day systems in nursing care, leading to the invisibility of oral health.

In the category “Accessibility and usefulness”, professionals expressed concerns regarding the relevant and sufficient information about the older adult’s oral status. The information on the oral care card needs to be easy to understand for all professionals; at the same time, different professions need to have different kinds of information. Based on the interviews, we suggest that the recommendations in the oral care card should be integrated with a common strategy, with the daily nursing care to increase visibility and pertinence.

There was no established placement for the oral care cards, and nursing staff in home care services were unable to tell where the oral care card was kept. The dental hygienists proposed that there is a risk of oral care being neglected when the oral care card is not placed in the bathroom, for example on the wall beside the bathroom sink. To the nursing staff, there is an ethical consideration regarding the placement of the paper-based oral care card and the personal integrity of the older adult. Nursing staff in home care services and dental hygienists wanted the oral care card to be a part of a digital tool. Based on this result, we suggest that a digital oral care card may contribute to both the request from professions and organizations for adapted information, while respecting the older adults’ integrity and providing nursing staffs access to the oral care card.

In the category “Coordination between nursing and dental care”, the differences between a nursing and dental care perspective emerged. Participants from both nursing and dental care emphasized the need for assessing information on the oral care card and the progress in oral status in different ways, but no participant was able to describe the entire oral care process. We suggest that a future oral care card needs to be established in the different organizations which will make it possible to regard different professionals’ perspectives and an overall view of care.

Staff in leading positions expressed a lack of knowledge regarding their responsibility concerning the oral care card. This indicates the lack of surrounding interorganizational structures for responsibilities and the lack of a common strategy and goals regarding the oral care cards. We believe that an oral care card could be anchored in the nursing and dental care context by including common goals and measurements that could make it easier to navigate the shared oral care responsibility. However, interorganizational perspectives regarding nursing care is poorly explored in research [31].

An “Ethical approach”, which was the third category, needs to be intertwined every step of the way in nursing [32] and dental care [33] and in the borderland between those two, when addressing the issue of oral care in general for frail older adults [34]. To involve, engage, and learn from the older adults in the implementation of their oral care, and to integrate it to the individual care implementation plan is not only ethical [34] but could also be seen as a part of person-centered care [35]. To strategically involve the older adults’ decision-making and abilities in the interprofessional work could also be considered to ensure healthy ageing: “the process of developing and maintaining the functional ability that enables well-being in older age” ([1] p. 4). Therefore, we suggest that the ethical dimensions of oral care within nursing should be more visible and developed in a future oral care card.

For the dental hygienists and the nursing home manager, it was common but not optimal to hand over the oral care cards and information to any present person in the staff. A situation was described where information was orally passed through several people and details were lost during the process, which can be an issue of patient safety. Another issue of patient safety may be information about the older adult’s health. We suggest setting up common/fixed routines for handover of both oral care cards from dental care to nursing care, and necessary health information from nursing care to dental care.

In Sweden, health and dental care organizations have different financial prerequisites, different medical record systems, and are in part regulated by different laws [36]. Health, nursing, and dental care organizations work with the same individuals but may experience challenges to collaborative work. The nursing staff highlighted the need to help the older adults maintain functional abilities as long as possible, which may interfere with the dental hygienists’ recommendations regarding assisted oral care to older adults with poor oral health. This example stresses the need for an interprofessional dialogue. Collaborative work can benefit from common goals, knowledge, and an understanding of each other’s context [37].

The oral care card was found to be an instrument in a complex setting. Oral health is an integrated part of general health but seems to exist in a borderland between nursing and dental care. Oral health may not yet have found its place in the nursing care context, but we believe that by anchoring the oral care card in the nursing care context, it has potential to do so. Further research should investigate how oral care cards could be developed and designed to support these goals.

### 4.2. Discussion—Method

Since no previous research was present concerning oral care cards in Sweden, we chose an inductive qualitative approach [38]. Elo et al. [39] presented a checklist for researchers regarding attempting the trustworthiness of a content analysis study. The checklist was an inspiration and added valuable input when discussing and conducting this study. To increase trustworthiness and credibility of the study [27], the analysis was conducted by two of the authors (JP and IJ) but discussed by all authors, who represented varied backgrounds. Participants with different professions were interviewed either as individuals or in group settings. We consider this to be a strength as the oral care card is related to several professions and we strived to obtain a broad picture of experiences and perceptions. Group interviews could lead to rich data when participants build on each other’s thoughts and experiences [40] if the study subject is not deemed as sensitive. Three interviews with persons in leading positions and a registered nurse were conducted individually due to the risk that status and hierarchical concerns could affect an open climate in group interviews [41].

The study was affected by the COVID-19 pandemic. The data collection started with face-to-face interviews, but the pandemic and public health recommendations caused the subsequent interviews to be digital. This mix of both physical and digital interviews was not planned but turned out well in terms of obtaining rich data and the possibility to reach participants from different geographical areas. In Sweden, there are many nursing assistants that are not permanently employed or do not have Swedish as native language. In this study, a larger part of the participants was permanently employed and had Swedish as native language (Table 1), which can be regarded as a limitation in terms of representativeness. However, using qualitative content analysis [23], the mixed sample of participants still led to a variation of experiences and perceptions. This made analysis complicated yet gave the broad picture of oral care work in nursing that we strived for. Further, Sandelowski and Leeman [42] argued that findings in qualitative research need to be understandable for the wider audience. To achieve this, the analysis was presented both in categories with subcategories at a concrete level as well as at a more latent level with a theme that concludes the analysis. Finally, in working with the manuscript, we followed the standards for qualitative research by O’Brien et al. [43].

## 5. Conclusions

The results indicate that experiences of the oral care card in the nursing care context are such that the oral health perspective may not yet have found its place. There is also a need for integrated interprofessional and interorganizational work regarding oral health in the context of the frail older adults. The results highlight the need for being able to tailor the oral care cards to different professions. Finally, ethical aspects regarding oral health of frail older adults, especially in staff-assisted oral care, is something that should be considered when professionals use the oral care cards.

## Figures and Tables

**Table 1 ijerph-19-09380-t001:** An overview of the distribution of participants according to type of interview, native language, and employment.

Group Interviews	*n*	Swedish as Native Language	Worked 6 Years or More	Permanently Employed
Group 1. Nursing assistants in LTC facility	6	6	5	6
Group 2. Nursing assistants and oral care aides in LTC facilities ^1^	3	2	3	3
Group 3. Nursing assistants in home care services	5	3	2	3
Group 4. Nursing assistants in home care services ^1^	2	1	2	2
Group 5. Nursing assistants in home care services ^1^	4	4	1	2
Group 6. Dental hygienists that perform oral assessments for oral care cards	5	5	5	5
Group 7. Municipal head of operations/strategist & a municipal dental hygienist ^1^	2	2	2	2
**Individual interviews**				
Registered nurse in LTC facility ^1^	1	1	0	1
Manager in LTC facility ^1^	1	1	1	1
Manager for oral screenings, dental services care ^1^	1	1	1	1
**Total**	30	26	22	26

^1^ Conducted digitally.

**Table 2 ijerph-19-09380-t002:** An example from the analysis regarding the category “Coordination between nursing and dental care”.

Condensed Units of Meaning	Codes	Subcategories	Categories
The nursing staff should be present during the oral assessments, but where does this responsibility lie?	Where does the responsibility lie?	Uncertain distribution of work and responsibilities.	Coordination between nursing-and dental care.

**Table 3 ijerph-19-09380-t003:** Overview of the results in theme, categories, and subcategories.

**Theme**	Navigating an Oral Care Responsibility That Is Not Anchored in Nursing and Dental Care Context		
**Category**	Accessibility and usefulness	Coordination between nursing and dental care	Ethical approach
** *Subcategories* **	Need of overview of dental status	Unstructured handover	Need to regard the state of the older adult
	Need of description of oral care needs	Uncertain distribution of work and responsibilities	Need to respect the older adult’s preferences
	Need of customization for different users	Lack of specific follow-up	Need to promote participation
	Need of oral care card to be within reach	Non-integrated systems	

## Data Availability

Participants of the study agreed for their data to be used. In agreement with GDPR, the data are only used for the purpose of this study.

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
