# Peer review of "Oral Care Cards as a Support in Daily Oral Care of Frail Older Adults: Experiences and Perceptions of Professionals in Nursing and Dental Care—A Qualitative Study"

_ijerph, 2022, doi:10.3390/ijerph19159380_

Round 1

Reviewer 1 Report

First of all, I would like to thank the authors for the effort they have made in researching this topic. It is a very interesting study presenting the experience and perceptions of professionals in nursing and dental care regarding oral care cards as a support in daily oral care of frail older adults. I personally encourage this type of study, as qualitative research takes an interpretive, naturalistic approach to its subject matter. 

However, I would like to recommend some considerations: 

Title- the type of study should be mentioned (qualitative study)

Introduction- correct

Materials and methods:

How did you establish the number of participants in the study (Sample size calculation).

What is the difference between group 4 and 5?

Age distribution of participants. 

Results: adequately presented

Discussion

Should discuss the choice for a qualitative instead of a quantitative study

Limitations of the study (proportion between native and non-native Swedish nursing staff, etc, use of quantitative analysis, number of participants, etc.). 

Conclusions

Erase : „Further research should investigate how oral care cards could be developed and designed in order to support these goals „- this does not represent a conclusion of the current research.

References

Should be formatted according to the MDPI indications

Author Response

Thank you for reviewing our manuscript. Please see the attachments. 

Reviewer 2 Report

The article entitled “Oral Care Cards as a Support in Daily Oral Care of Frail Older Adults: Experiences and Perceptions of Professionals in Nursing and Dental Care” aimed to explore nursing and dental professionals’ experiences and perceptions of oral care cards.

The paper is in line with journal’s aim, moreover, Authors have well revised several issues; however, I ask authors to add aimed to provide some key concepts.  

-       In the introduction section the topic related the need of support

-       systems in healthcare that consider an aging population with complex needs of care and the quote on the 2030 agenda is interesting. It would also be more pertinent to discuss these objectives in the context of the pandemic era we are experiencing, and to evaluate how the Sars-Cov-2 pandemic has influenced dentistry and the approach to disabled and frail patients (please see and discuss DOI 10.3390 / healthcare9040454 and)

-       In the discussion the authors should evaluate and discuss the current state of the art on the management of disabled and frail patients in the clinic (please see and discuss DOI

-       10.3390 / ijerph18041556)

-       Conclusions cannot be reduced to a sentence: you must improve them highlighting the limits and the future insights pointed out from this article.

-       The formatting of the references is not correct, please check the journal instructions for authors

-       Several moderate typos are present in the text, please, amend

According to this Reviewer’s consideration, novelty and quality of the paper, publication of the present manuscript is recommended after minor revision.

Author Response

Thank you for reviewing this manuscript. Please see the attachment. 

Reviewer 3 Report

The introduction was well written by providing information regarding importance of study, health care system in Sweden and explaining oral health care card usage system. This provides clear justification for the study while providing sophisticated background for a non-Sweden reader.

Material and methods

Please provide justification for purposeful sampling.

Different parts of region – Not clear. Why were those parts selected? What regions?

Why oral care aides interviewed in a separate group? Please explain the table more.

Why 30 people were interviewed? Why not less or more? How the number was decided?

How was the trustworthiness of the data confirmed? Please add more information in methods and materials.

How other components of validity, reliability and generalizability confirmed?

What was the experience of interviewer on qualitative interview techniques? 

Any incentives provided? 

In what language was interview conducted? It is not clear. If English, what was the English proficiency level of the participants? If Swedish, how nonnatives were involved? If conducted in Swedish, who translated the interviews? How was the quality assured for translation?

Was there an option to conduct the interview in mother tongue provided?

How were the interviews recorded? Please elaborate the informed consent procedure.

Who transcribed the data? Professionals or researchers? How transcripts were checked for the quality?

The validity and generalizability of quantitative data can be provided to readers easily using the sampling procedure, effect size, significance etc. However, the qualitative data needs to be written elaborately to increase the trustworthiness of data. The justification provided for trustworthiness and other validity component are not enough. Please improve the methods and materials section. Otherwise, this is a very important study for our community.

Results and Discussion: Please emphasize the quote examples. It is not clear whether the examples are in raw format or edited by the authors. 

Thanks for providing an opportunity to review this study. This is a great study.

Author Response

(The authors gave the same response as above.)

Round 2

Reviewer 1 Report

The authors have modified the manuscript according to the suggestions and I consider it is now suitable to be published. 

Author Response

Thank you for your comments, please see the attachment. 

Reviewer 3 Report

Thanks for improving the paper carefully. I would like authors to double check whether the results are reported following the Standard for Reporting Qualitative Research (O’Brien et al., 2014) and mention in the methods section that the paper was reported following SRQR.

Author Response

(The authors gave the same response as above.)
